# The Virtual Patch Clamp: Imputing *C. elegans* Membrane Potentials from Calcium Imaging

**Andrew Warrington**
Department of Engineering Science
University of Oxford
`andrew.warrington@keble.ox.ac.uk`

**Arthur Spencer**
School of Physiology,
Pharmacology & Neuroscience
University of Bristol

**Frank Wood**
Department of Computer Science
University of British Columbia

## Abstract

We develop a stochastic whole-brain and body simulator of the nematode round-worm *Caenorhabditis elegans* (*C. elegans*) and show that it is sufficiently regularizing to allow imputation of latent membrane potentials from partial calcium fluorescence imaging observations. This is the first attempt we know of to "complete the circle," where an anatomically grounded whole-connectome simulator is used to impute a time-varying "brain" state at single-cell fidelity from covariates that are measurable in practice. Using state of the art Bayesian machine learning methods to condition on readily obtainable data, our method paves the way for neuroscientists to recover interpretable connectome-wide state representations, automatically estimate physiologically relevant parameter values from data, and perform simulations investigating intelligent lifeforms *in silico*.

## 1 Introduction

One of the goals of artificial intelligence, neuroscience and connectomics is to understand how sentience emerges from the interactions of the atomic units of the brain, to be able to probe these mechanisms on the deepest level in living organisms, and to be able to simulate this interaction *ad infinitum* [1]. In this work, we assemble an anatomically grounded, interpretable probabilistic brain-body simulator for the widely studied nematode roundworm *Caenorhabditis elegans* (*C. elegans*) [2, 3]. We then present methods for performing posterior inference in the time evolution of the state of the worm and estimate the global simulator parameter values from readily obtainable *non-invasive* calcium fluorescence data [4]. We refer to using an anatomically grounded model to infer latent states and parameters, conditioned on partial data, as a "virtual patch clamp" (VPC). The VPC also facilitates *in silico* experimentation on "digital" *C. elegans* specimens, by programmatically modifying the simulator and observing the resulting simulations; enabling rapid, wide-reaching, fully observable and perfectly repeatable exploration of hypotheses into the how the fundamental units of the neural circuit of *C. elegans* combine to create intelligent behaviour.

## 2 Simulating *C. elegans*

Due to the simplicity and regularity of its anatomy, and its predictable yet sophisticated behavioural repertoire, *C. elegans* is used as a "model organism" across biology and neuroscience research. Notably, its connectome is regular across wild-type specimens and has been mapped at synapse and

33rd Conference on Neural Information Processing Systems (NeurIPS 2019), Vancouver, Canada.

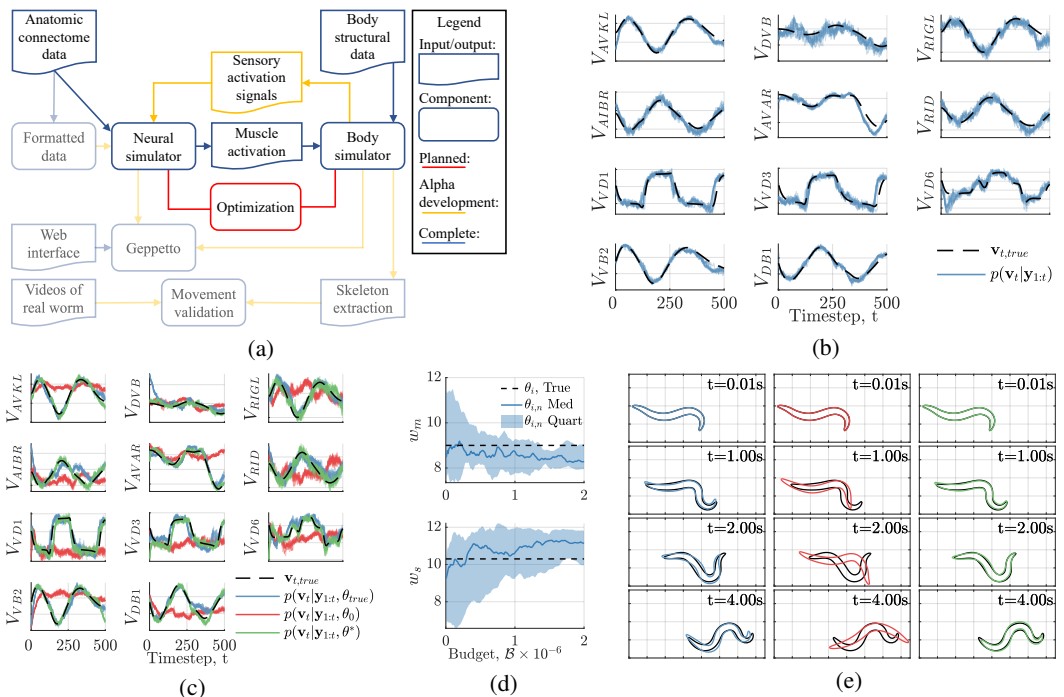

Figure 1: (a): Diagram adapted from Sarma et al. [1] reflecting the community planned development pipeline for *C. elegans* simulation. Greyed out components are not considered in this paper. The status of components are as categorized by OpenWorm [1]. (b): Successful recovery of latent states conditioned on just 49 calcium traces, where the true trajectory is shown in black, as introduced in Section 4 (c), (d), and (e): Learning better parameters, as introduced in Section 4 and shown in (d) enables better recovery of latent voltage trajectories and body states (blue, Subfigures (c) and (e) respectively) compared with parameters drawn from the prior (red). This also indicates that the quality of the posterior predictive distributions and unconditional generations are of higher quality.

gap junction fidelity using electron microscopy [2, 3]. Because of this fixed architecture, neural circuit simulators, imbued with anatomically correct structure, have been developed to produce feasible whole *C. elegans* connectome simulators [5] by leveraging highly accurate neural dynamics models [6]. Likewise, its simple anatomy has allowed body and locomotion simulators to be developed [7]. The first contribution of this paper is a new *C. elegans* simulator that integrates existing simulators and models [5, 7, 8] developed by the *C. elegans* community. At a high level, our simulator is comprised of three components: a simulator for the time-evolution of the membrane potential [5] and intracellular calcium ion concentration [8] in *all* 302 *C. elegans* neurons, a simulator for the physical form of the worm and the associated neural stimuli and proprioceptive feedback [7], and a model relating the intracellular calcium to the observable florescence data [4, 8].

The first component of our model is a simulator of connectome-scale, single-neuron fidelity neural dynamics. We modify the simulator presented by Marblestone [5], which builds on Wicks et al. [6], called 'simple *C. elegans*' (SCE). SCE is designed to be an easily interpretable simulator of *C. elegans* neural membrane potential dynamics via single-compartment neuron models connected by chemical synapses and electrical gap junctions. Exemplar voltage traces generated by our simulator are shown as black dashed lines in Figure 1(b). We add to SCE a model for intracellular calcium ion concentration [8]. We also integrate a simulator of the body shape of the worm, WormSim [7]. WormSim models the body shape in two dimensions as a series of rods, contractile units and springs driven by impulses generated by a simplified neural network. We integrate the anatomically correct representation used by SCE to drive WormSim and receive proprioceptive feedback. A typical evolution of body state is shown in black in Figure 1(e). Finally we incorporate a model of the fluorescence signals observed through calcium imaging. This dependence is described by a saturating Hill-type conditioned on intracellular calcium concentration [8], where only $M$ of the 302 neurons are observed and identified (here $M = 49$, see Kato et al. [4]).

To summarize our model, the neuron states, body state, and proprioceptive feedback define the latent "brain" and "body" state of the worm, denoted at time $t$ as $\mathbf{x}_t \in \mathbb{R}^{994}$. The observed data, $\mathbf{y}_t \in \mathbb{R}_+^M$, is the calcium florescence signal. We now demonstrate how the tools of Bayesian inference can be employed to condition simulations on partial observations, make predictions conditionally or unconditionally, and perform marginal maximum a posteriori parameter estimation.

## 3 The Virtual Patch Clamp

The second contribution of this paper is the adoption and scaling of a method to impute the entire latent state, $\mathbf{x}_t$, conditioned on observable calcium imaging florescences. We wish to quantify the distribution over the latent states conditioned on the observed data, referred to as the posterior distribution $p(\mathbf{x}_{0:T}|\mathbf{y}_{1:T}, \boldsymbol{\theta})$. To relate how this achieves to our outlined objectives, this represents, under the model, the distribution over all latent neural and physiological states, $\mathbf{x}_t$, conditioned on the observed data, providing the imputation element of the VPC. Forward simulation of the particles initialized from the posterior distribution at $T$ provides posterior predictive inference over state evolution, where, for instance, physiological variables can be programmatically clamped (inspiring the name VPC). Finally the posterior, $p(\boldsymbol{\theta}|\mathbf{y}_{1:T}) = p(\mathbf{y}_{1:T}|\boldsymbol{\theta})p(\boldsymbol{\theta})$, allows us to objectively compare models and hypotheses, which will be used later for parameter estimation. Due to the non-invertible, non-differentiable nature of the simulator, we use sequential Monte Carlo (SMC) for estimating the posterior as a weighted discrete measure approximating the target distribution, as well as providing an estimation of the model evidence, $p(\mathbf{y}_{1:T}|\boldsymbol{\theta})$ [9].

In our first experiment we first generate a synthetic state trajectory by sampling from the model, and then recover the known ground-truth trajectory from observed fluorescence traces using a fixed model. Specifically we condition on the same 49 neurons identified in the calcium imaging data released by Kato et al. [4]. Results for this are shown in Figure 1(b), where the true state is shown in black, while the filtering distribution recovered by SMC is shown in blue.

The blue reconstructions are congruent with the black trace, indicating that the latent behaviour of the complete system is being well-reconstructed despite partial observability. Critically, neurons not directly connected to observed neurons (for instance VD6) are correctly reconstructed, indicating that the regularizing capacity of the model is sufficient to constrain these variables. Further confirmation of the power of this method can be seen in the leftmost column of Figure 1(e), showing the predicted body shape closely matches the true state, despite not being explicitly conditioned upon body shape. This experiment shows that the VPC is tractable and is capable of yielding high-fidelity reconstructions of pertinent latent states given partial calcium imaging observations via the application of Bayesian inference to time series models of *C. elegans*.

## 4 Parameter Estimation

The posterior inference and evidence approximation presented in the previous section is useful for imputing values and performing *in silico* experimentation. In the previous section we fixed the model to demonstrate the viability of SMC for latent state imputation. We now allow the parameters of the simulator, collectively denoted $\boldsymbol{\theta}$, such as the non-directly observable electrical and chemical characteristics of individual synapses in the *C. elegans* connectome, as well as parameters of the body model, the calcium fluorescence model, etc, to be unknown and hence must be learned. We conclude this paper by taking concrete steps towards performing such parameter estimation, as defined by the simulator-structured hypothesis class defined by the chosen model.

Our goal is to estimate the best simulator parameters $\boldsymbol{\theta}^*$ given observed data, i.e. $\boldsymbol{\theta}^* = \mathrm{argmax}_\theta \, p(\boldsymbol{\theta}|\mathbf{y}) = \mathrm{argmax}_{\boldsymbol{\theta}} \, p(\mathbf{y}|\boldsymbol{\theta})p(\boldsymbol{\theta})$. The method we employ for performing parameter estimation is a novel combination of variational optimization (VO) [10] and SMC evidence approximation. This results in a stochastic gradient for parameter estimation that does not require a differentiable simulator and can deal with a large number of latent variables. VO starts with the following bound [10]

$$\min_{\boldsymbol{\theta}} f(\boldsymbol{\theta}) \leq \mathbb{E}_{\boldsymbol{\theta} \sim q(\boldsymbol{\theta}|\boldsymbol{\phi})}[f(\boldsymbol{\theta})] = U(\boldsymbol{\phi}). \tag{1}$$

The gradient of $U(\boldsymbol{\phi})$ with respect to $\boldsymbol{\phi}$ can then be computed as

$$\nabla_{\boldsymbol{\phi}} U(\boldsymbol{\phi}) = \nabla_{\boldsymbol{\phi}} \mathbb{E}_{\boldsymbol{\theta} \sim q(\boldsymbol{\theta}|\boldsymbol{\phi})}[f(\boldsymbol{\theta})] = \mathbb{E}_{\boldsymbol{\theta} \sim q(\boldsymbol{\theta}|\boldsymbol{\phi})}[f(\boldsymbol{\theta}) \nabla_{\boldsymbol{\phi}} \log q(\boldsymbol{\theta}|\boldsymbol{\phi})], \tag{2}$$

where Monte Carlo integration is used to evaluate this expectation.

The objective function is the joint density $f(\boldsymbol{\theta}) = -p(\mathbf{y}, \boldsymbol{\theta}) = -p(\mathbf{y}|\boldsymbol{\theta})p(\boldsymbol{\theta})$, where the likelihood term is approximated via SMC. To our knowledge, this is the first time that pseudo-marginal methods have been paired with variational optimization methods. We refer to this procedure as particle marginal variational optimization (PMVO). We implement a framework for embarrassingly parallel evaluation of multiple SMC sweeps on large, distributed high performance compute clusters, where each SMC sweep is executed on a single node, eliminating network overheads.

We conclude by demonstrating the utility of our PMVO technique by recovering known simulator parameters on synthetic data generated by the model. For this work, we optimize the two parameters we introduced by integrating SCE and WormSim, namely the strength of motor stimulation, $w_{\mathrm{m}}$, and proprioceptive feedback, $w_{\mathrm{s}}$. The results of this experiment are shown in Figure 1. Figures 1(c) and 1(e) show the imputed voltage traces and body poses when using the true parameters (blue), initial parameters (red) and optimized parameters (green), conditioned on just $49$ neurons. Recovery of "good" parameter values facilitates good imputation of latent states, especially for body position which is not explicitly conditioned on after initialization. Figure 1(d) shows the distribution of convergence of the two parameters towards the true value. This experiment shows that parameter inference in *C. elegans* models using PMVO is viable. Increasing the number of particles used in the SMC sweeps, the number of samples drawn from the proposal and observing more neurons (although currently logistically infeasible) improves the quality of the reconstructions and recovery of parameters.

## 5 Discussion

In this work we have explored performing Bayesian inference in whole-connectome neural and whole-body *C. elegans* simulations. We describe the model-based Bayesian inference aspect of this as a "virtual patch clamp," whereby unobserved latent membrane potentials can be inferred from partial observations gathered non-invasively. Our choice of inference method facilitates estimation of the model evidence, a measure of how well the model explains the observed data. We presented a method for maximizing this evidence without requiring differentiable simulation components. In the past year several articles discussing open research issues pertaining to *C. elegans* simulation have been produced by the *C. elegans* community [1, 11]. Figure 1(a) outlines the community planned development pipeline for *C. elegans* simulation. Our work addresses the implementation of the box simply labelled "optimization." We show on representative synthetic data that our method is capable of performing such an optimization. This approach promises to allow neuroscientists to peer deeper into the neural function of a living organism, testing hypothesis on neural function that were previously unreachable. It is widely touted that convolutional neural networks were developed by wide-scale study of the V1 cortex. We believe connectome-level optimization and simulation, as demonstrated here, is the next step in neuroscience to understanding the very root of intelligence, but also discovering and developing techniques building towards artificial general intelligence.

## 6 Acknowledgements

Andrew Warrington is funded by the Shilston Scholarship, Keble College, Oxford. Arthur Spencer is supported by the Wellcome Trust. We acknowledge the support of the Natural Sciences and Engineering Research Council of Canada (NSERC), the Canada CIFAR AI Chairs Program, Compute Canada, Intel, and DARPA under its D3M and LWLL programs.

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
