# OpenReview forum: "The Virtual Patch Clamp: Imputing C. elegans Membrane Potentials from Calcium Imaging"
_NeurIPS.cc/2019/Workshop/Neuro_AI — Real Neurons & Hidden Units @ NeurIPS 2019 Poster_

### Official Review · AnonReviewer2 · 2019-09-21
**Promising approach for in silico neuroscience**

**Clarity:** 3

**Comment:**

Interesting work using modeling and measured data to build tools for better interrogating biological neural networks in simulation. The model fits could be better quantified, and the relationship between this work and other existing c. elegans simulations that exist would be beneficial.

**Category:**

AI->Neuro

**Clarity Comment:**

I find the figure and associated legend somewhat unclear. The legend itself does not describe what is presented in the figures--instead that is done within the text itself. The manuscript also discusses panels within the figure out of order a bit. As someone outside of the c. elegans community, I also found some of the presentation of details lacking (e.g. how calcium data specifically are translated into constraints on the model). Some brief description of details could be useful to widen the relevant audience. Similarly, I know there are many other existing models/simulators of C. Elegan nervous systems. Some introduction or discussion to put this work in context with that existing literature would be beneficial.

**Evaluation:**

4: Very good

**Importance:**

3: Important

**Importance Comment:**

The author(s) develop strategies to use partial measurements to constrain and fit whole-brain + body simulations of a c. elegan. The resulting models would be useful tools of significant interest for neuroscience.

**Intersection:**

3: Medium

**Intersection Comment:**

The work represents a nice example of improved modeling of a nervous system based on measured data. This is broadly of interest for neuroscience research, however the relevance of the work for machine learning/AI is not well-articulated in the manuscript.

**Rigor Comment:**

The manuscript could be made much more convincing by incorporating more efforts to quantify degree of model fits beyond qualitative assessment presented.

**Technical Rigor:**

2: Marginally convincing

---

### Official Review · AnonReviewer3 · 2019-09-25
**Full state inference from partial calcium fluorescence recordings in C. elegans that predict body position in a simulator**

**Clarity:** 4

**Comment:**

Strengths:

A thorough and interesting new approach to the problem.

Areas for improvement:

Quantification of reconstruction quality. I would also like to know what the alternatives were if you didn't develop your PMVO method, and why they would or would not have worked as well.

**Category:**

AI->Neuro

**Clarity Comment:**

The problems and solutions are clearly described.

**Evaluation:**

4: Very good

**Importance:**

4: Very important

**Importance Comment:**

The authors describe a new method for variational inference with pseudo-marginal likelihoods that can deal with a large number of variables and does not require a differentiable simulator. This could also have a lot of application to other biologically realistic life science simulations. They apply it to inferring the full state of the nervous system of C. Elegans from partial observation.

**Intersection:**

4: High

**Intersection Comment:**

This paper develops a new optimization technique for inferring nervous system state, so is clearly at the AI-Neuro intersection.

**Rigor Comment:**

The approach describes a variational method to infer the latent state of the nervous system of C. Elegans from observation of a subset of the cells via calcium imaging. They show that they are capable of reconstructing the latent state, and are also capable of repoducing Worm behaviour. Visually these are convincing, however, they do not show any quantification of the quality of reconstructions.

**Technical Rigor:**

4: Very convincing

---

### Official Review · AnonReviewer1 · 2019-09-27
**Intriguing brain-body simulation of C. elegans but contributions unclear**

**Clarity:** 2

**Comment:**

Promising study that would benefit from clarifications.

**Category:**

AI->Neuro

**Clarity Comment:**

Abstract unclear about the achievements:
“ to allow imputation of latent membrane potentials from partial calcium fluorescence imaging observations”.
=> What does it mean?
“Using state of the art Bayesian machine learning methods”
=> Which state of the art methods?
“simulations investigating intelligent lifeforms in silico”
=> What properties of lifeforms are being investigated precisely?

Intro
“be able to simulate this interaction ad infinitum” => what kind of infinity?

Sec 3:
"Critically, neurons not directly connected to observed neurons (for instance VD6) are correctly reconstructed, indicating that the regularizing capacity of the model is sufficient to constrain these variables."
=> Was this observed neuron not used as one of the calcium traces for inferring the model? What are the latent states? Unclear what part of the data is predicted and what part is observed.

Fig 1 caption:" Successful recovery of latent states conditioned on just 49 calcium traces, where the true trajectory is shown in black, as introduced in Section 4 "
=> What are the latent states?

Bayesian inference method: "The method we employ for performing parameter estimation is a novel combination of variational optimization (VO) [10] and SMC evidence approximation. To our knowledge, this is the first time that pseudo-marginal methods have been paired with variational optimization methods. "
=> Difficult to evaluate the importance of this new combination for the field (Disclaimer: I am not an expert).





**Evaluation:**

3: Good

**Importance:**

3: Important

**Importance Comment:**

I am not expert in C-elegans models so it is hard to evaluate the contributions of this paper and their novelty. I however have concerns about the clarity of the exposition of the findings.



**Intersection:**

3: Medium

**Intersection Comment:**

Interesting for neuroscience, less clear for AI.

**Rigor Comment:**

The works seems rigorous but some clarity issues prevent good evaluation.

**Technical Rigor:**

3: Convincing

---

### Decision · Program_Chairs · 2019-10-02

Accept (Poster)